# Diabetes and physical activity: A prospective cohort study

H. M. Dumidu A. B. Attanayake[1], Adrian Barnett[2,3], Nicola W. Burton[4,5], Wendy J. Brown [6], Susanna M. Cramb [2,3,7]*

**1** Group Human Resources, MAS Capital (Pvt) Ltd, Colombo, Sri Lanka, **2** School of Public Health and Social Work, Queensland University of Technology, Brisbane, Queensland, Australia, **3** Australian Centre for Health Services Innovation & Centre for Healthcare Transformation, Queensland University of Technology, Brisbane, Queensland, Australia, **4** School of Applied Psychology, Griffith University, Brisbane, Queensland, Australia, **5** Menzies Health Institute Queensland, Griffth University, Brisbane, Queensland, Australia, **6** School of Human Movement and Nutrition Sciences, University of Queensland, Brisbane, Queensland, Australia, **7** Centre for Data Science, Queensland University of Technology, Brisbane, Queensland, Australia

\* susanna.cramb@qut.edu.au

## Abstract

Diabetes is on the rise as the worldwide population ages. While physical activity can help protect against diabetes, ageing is commonly associated with reduced physical activity. This study aimed to examine if physical activity differs by diabetes status in mid-aged adults, how this association changes over time, and whether physical activity-related sociodemographic factors and health indicators differ in those with and without diabetes. Data came from four waves of the How Areas in Brisbane Influence HealTh and AcTivity (HABITAT), a longitudinal study of mid-age adults living in Brisbane, Australia. Random effects/Expectation-maximisation (RE-EM) regression trees were used to identify factors affecting physical activity among those with and without diabetes, both separately and combined. At study entry, those with diabetes had a higher median age of 58 years (95% CI: 57–60) and a lower median physical activity of 699 MET.min/week (95% CI: 599–799) than people without diabetes (53 years (95% CI: 53–53) and 849 MET.min/week (95% CI: 799–899)). However, the strongest factors influencing physical activity were BMI and gender, not diabetes status. It is vital to promote physical activity among adults, in particular among those with high BMI and women, as well as those with and at high risk of diseases like diabetes.

## Introduction

Diabetes causes a substantial and increasing disease burden worldwide. An estimated 463 million people (9.3% of the world's population) have diabetes, half of whom are unaware of their condition [1]. By 2045 this is expected to be 700 million (10.9%) [1]. In Australia, one in twenty (4.9% or 1.2 million) people have a diagnosis of diabetes [2], most of which is type 2.

Diabetes prevalence tends to increase with age. In Australia, 2% of people aged 35 to 44 years have been diagnosed with diabetes, and this increases to 19% for those aged 75+ years [2]. As the population in Australia ages, consistent with many other developed countries, the number of people with diabetes is likely to increase.

been moved from the institution from which ethics approval was obtained. Researchers interested in collaborating with the HABITAT Chief Investigators and accessing the data should contact the Lead Investigator, Professor Gavin Turrell (gavin. turrell@rmit.edu.au).

**Funding:** The HABITAT study was supported by the Australian National Health and Medical Research Council (NHMRC; https://www.nhmrc.gov.au/) grants #1047453, #497236, and #339718, with support from the Brisbane City Council. SC was funded by an NHMRC Investigator grant (#2008313). The funders had no role in study design, data collection and analysis, decision to publish, or preparation of the manuscript.

**Competing interests:** The authors have declared that no competing interests exist.

Physical activity is known to be beneficial in reducing diabetes risk [3], but often declines as people age [4]. Many sociodemographic and health factors can adversely influence physical activity, including neighborhood disadvantage [5] and low socioeconomic status [6, 7], mental health issues such as depression [8, 9], and high body mass index [10]. It is unclear whether having diabetes influences physical activity levels [11].

The aim of this study was to explore whether physical activity differs by diabetes status in mid-aged adults, how this association changes over time, and whether relevant sociodemographic factors and health indicators differ in those with and without diabetes.

## Method

### Study design and data collection

This project used data from four (2009, 2011, 2013 and 2016) of the five waves in the How Areas in Brisbane Influence HealTh and AcTivity (HABITAT) longitudinal study of mid-age adults living in Brisbane, Australia which commenced in 2007 [12, 13]. Since measures of interest for the current study were first collected at wave 2 in 2009, this was used as the baseline. Ethics approval for the HABITAT study was obtained from the Human Research Ethics Committee at Queensland University of Technology (Ref. no. 3967H).

HABITAT was designed to track physical activity and explore associated environmental, social, psychological and socio-demographic factors. A multistage probability sampling design was used with a stratified random sample (n = 200) of small geographical areas known as Census Collection Districts (CCDs). In each CCD a random sample of residents (85 people on average) aged 40 to 65 years were selected, and a paper survey mailed. The number of eligible participants for the five waves (2007 to 2016) were 16,127, 10,828, 10,209, 9651, and 8826 and the percentage of valid responses received in each wave were 68%, 73%, 68%, 68% and 59%, respectively [13].

### Outcome variable

We used self-reported physical activity which was assessed as time spent during the previous seven days in each of: continuous walking for exercise or transport, moderate intensity activity, and vigorous activity (see S1 Appendix for questionnaire). To minimize potential over-reporting, the time for each activity was truncated at 14 hours [12], as is standard protocol. Total physical activity (MET.min/week) was derived as follows [14]:

$$\text{physical activity score (MET.min/week)}$$
$$= \text{walk minutes} \times 3.33 + \text{moderate minutes} \times 3.33 + \text{vigorous minutes} \times 6.66$$

A score above 500 MET.min/week reflects $\geq$ 150 minutes of walking or moderate intensity activity, or $\geq$ 75 minutes of vigorous activity, or a combination of walking, moderate and vigorous activities, commensurate with the lower end of the range recommended in the current Australian [14] and WHO [15] guidelines.

### Exposure variable

Diabetes was assessed in response to the question "Have you ever been told by a doctor or nurse that you have any of the LONG-TERM health conditions listed below? Please only include those conditions that have lasted, or are likely to last, for six (6) months or more." (Followed by a list of 18 conditions, including diabetes). This question has been extensively used in Australian research [2], and self-reported measures for chronic conditions have been demonstrated to be reliable [16].

We categorized each respondent as being with or without diabetes, considering their responses across all study waves. Any individual identified as consistently having diabetes was considered to have diabetes, while individuals who consistently stated they did not have diabetes were considered to not have diabetes. Individuals who changed their diabetes status between waves were excluded.

## Independent variables

Seven sociodemographic, health and wellbeing variables were chosen, based on previously demonstrated evidence of an association with either diabetes [17–19] and/or physical activity [7, 20, 21] (S1 Fig and S1 Table). Gender and education data were obtained from 2007 survey data and fixed over the four waves, while values of the other variables varied over time. Any missing categories of these variables were also included in the analysis since the RE-EM tree algorithm can handle missing values in the independent variables.

1. Gender as male or female.

2. Education. The nine response options for highest education qualification were recoded as: (1) No Post School Qualification (Year 9 or less, Year 10, Year 11, Year 12); (2) Vocational Qualification (Certificate); (3) Diploma (Diploma/Associate degree); (4) Bachelor's degree or Higher (Bachelor's degree, Graduate Diploma/Graduate Certificate, Postgraduate degree).

3. BMI category at each wave: respondents self-reported their height and weight. BMI was calculated as weight (kg) / [height (m)]$^2$ and classified according to the standard BMI categorization [22] of: (1) Underweight (BMI < 18.5); (2) Normal Weight (BMI: 18.5–24.9); (3) Overweight (BMI: 25.0–29.9); (4) Class 1 obesity (BMI: 30.0–34.9); (5) Class 2 obesity (BMI: 35.0–39.9); (6) Class 3 obesity (BMI 40+).

4. Psychological distress at each wave: the Kessler-6 [23] asked whether in the last 4 weeks respondents had felt: Nervous; Hopeless; Restless or fidgety; So sad that nothing could cheer you up; That everything was an effort; Worthless. Possible responses were: None, A little, Some, Most, All of the time; with values ranging from 1 (None) to 5 (All). Items were summed with the minimum score possible as 6, and maximum 30. Each individual was categorized into one of four psychological distress groups:

   1. Low (or no) psychological distress (Kessler value: 6–9),

   2. Moderate psychological distress (Kessler value: 10–13),

   3. High psychological distress (Kessler value: 14–17),

   4. Very high psychological distress (Kessler value: 18–30).
      These four groups were based on those used by the Australian Bureau of Statistics to categorize the Kessler-10 scale [24].

5. Annual gross household income (AUD) at each wave: The 13 response options were recategorized as: (1) < $25,999, (2) $26,000–51,999 (3) $52,000–72,799, (4) $72,800–129,999, (5) ≥ $130,000, (6) Don't Know/ Don't want to answer this, as for previous analyses of HABITAT data [25].

6. Age in years at each wave were calculated using the date of birth from the 2007 wave.

7. Neighborhood disadvantage at each wave: Derived using weighted linear regression from the Australian Bureau of Statistics' Socioeconomic Indexes for Areas using the Index of

Socioeconomic Disadvantage [26] in each census collection district or the smaller statistical area 1 (from 2011 onwards) for each census during 1986, 1991, 1996, 2001, 2006, 2011 and 2016, linear trends were calculated to determine a value for each year and neighborhood. These were provided as percentiles of Socioeconomic Disadvantage for each HABITAT neighborhood (similar in size to the census collection district) and survey wave, ranging from 1 (very disadvantaged) to 100 (least disadvantaged).

## Statistical analysis

Of the 11,035 respondents in the HABITAT initial survey (year 2007) who reported physical activity, we excluded 535 who withdrew before wave 2, and 166 respondents who were considered unlikely to be the same respondent over all waves (Fig 1). A further 531 respondents who reported requiring assistance with tasks of daily living due to a long-term disability or illness at any wave were excluded because their ability to engage in physical activity may have been limited. For the remaining respondents, observations at waves which did not include physical activity data were removed (8,248 observations). Most of these observations missing physical activity values had either not returned a valid survey in that wave, or had withdrawn from the study before the wave.

The resulting analytical sample considered to have diabetes was 362 people (960 observations) and the number consistently without diabetes was 7,338 (21,768 observations).

## Modelling strategy

Random effects/Expectation-maximisation (RE-EM) regression trees [27] were used as these models allow for repeated data from individuals, time-varying covariates (e.g. BMI), and unequal time periods between waves. These combine the structure of a mixed effects linear model, often used when an individual is monitored over time, with the flexibility of tree-based estimation methods [28]. Tree-based methods are useful for non-linear associations and when covariates interact with each other. Starting at the root node, a tree is grown using binary splits until a stopping rule is reached [29]. The predicted value in each terminal node (also known as 'leaves') is the node sample mean. While all covariates are entered into the tree model, ideally only the covariates that are useful in distinguishing between observations remain in the output tree. The calculation alternates between estimating the regression tree while assuming the random effects are correct, and estimating the random effects assuming the regression tree is correct.

RE-EM can outperform standard tree methods for longitudinal data [28]. In tree-based methods, the nodes may split on any covariate, so different observations for the same individual may be placed in different nodes. Covariates may be constant over time, constant across individuals, or varying across time and individuals. Missing covariate information can also be included, but the outcome (here, physical activity) must not be missing. Missing covariates were handled using surrogates that assume the data are "missing at random" as surrogates use correlations in the observed data.

Further details are provided in S1 Appendix.

A sensitivity analysis further considered the sampling design and modified the random effects to have individuals nested within baseline census collection districts. However, this had negligible impact likely due to the moderate influence of census collection district after accounting for the other variables, so results are presented only for the initial model.

A key difference between this tree estimation and the more popular CART algorithm is that the CART model overfits then prunes using cross-validation, whereas the RE-EM tree has an

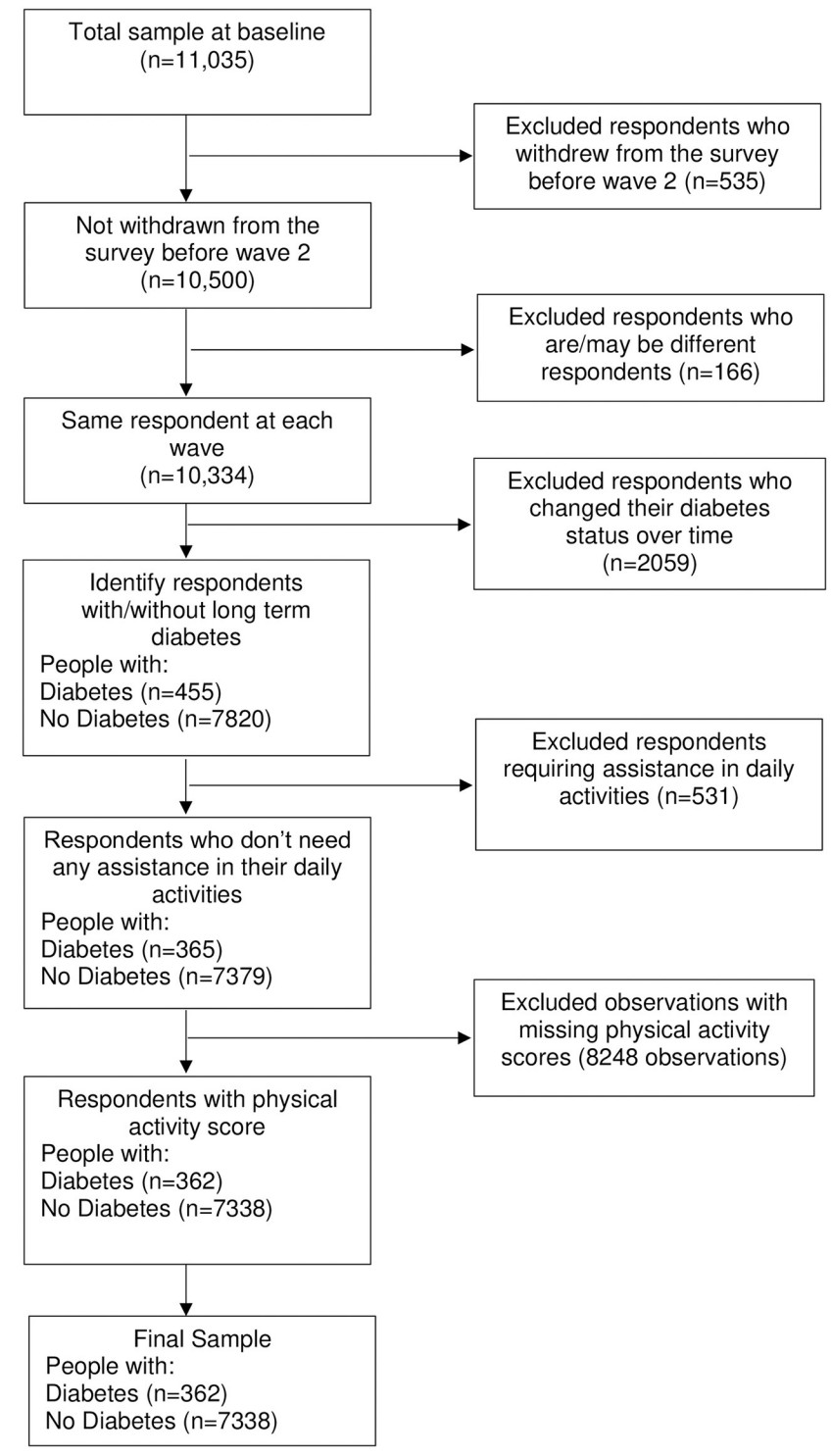

**Fig 1. Process Chart for the study.**

early stopping approach [30]. The sample R code provided for RE-EM unbiased tree construction with the REEMctree function [27] was the basis for our analysis.

**Predictive performance of the trees.** To evaluate the prediction performance of the trees, the predicted root mean square error (PMSE) per individual was used, calculated as:

$$\text{Root PMSE} = \sqrt{\frac{\sum_{i=1}^{I}\sum_{t=1}^{T}(y_{it} - \widehat{y_{it}})^2}{IT}}$$

Here $i$ refers to individuals, $t$ refers to time points, $y$ is the actual value while $\hat{y}$ is the corresponding predicted value. Smaller values indicate a closer fit to the data.

To further help understand model prediction, box plots were used to show the distribution of observed physical activity scores in each predicted terminal node (S2–S4 Figs).

For cross validation purposes 80% of the non-diabetes population considered for the study were used as the training data set and the remaining 20% was the testing data set. The same process was repeated with all data included. Results shown are for all data combined.

All data analyses were conducted in R software 4.0.4 [31].

## Results

The median age of all respondents in 2009 was 53 years (95% CI:53–54), and the median physical activity score was 833 MET.min/week (95% CI:799–899). Table 1 summarizes the characteristics of the analytic sample at study entry by diabetes status. The 362 respondents with diabetes had a higher median age of 58 years (95% CI: 57–60) and a lower median physical activity of 699 MET.min/week (95% CI: 599–799) than people without diabetes (53 years (95% CI: 53–53) and 849 MET.min/week (95% CI: 799–899), respectively). Physical activity scores among respondents with diabetes tended to be slightly lower at each wave than among individuals without diabetes (Fig 2a). Although individual trajectories could greatly vary, physical activity declined slightly over time among those with diabetes (Fig 2b). At wave 2, people with diabetes had a higher physical activity score (mean: 1343; median 716 MET.min/week) than at wave 5 (mean: 1131; median 599). Despite this, BMI overall remained consistent and markedly higher among people with diabetes over time, while psychological distress tended to slightly reduce between waves 2 and 5 (Fig 2c & 2d).

### Physical activity among respondents with diabetes

The strongest predictor of physical activity among respondents with diabetes was gender (Fig 3). Men had a higher average physical activity than women. Among women, those with at least a bachelor qualification had, on average, almost double the physical activity of those with lower levels of education.

### Physical activity among respondents without diabetes

The strongest predictor of physical activity among respondents without diabetes was BMI. Predictors in decreasing order of strength were then gender, neighborhood disadvantage, psychological distress, income and education. Due to the many variables involved, results are shown in two parts based on the first split: normal to overweight BMI categories (Fig 4a) and obese BMI categories (Fig 4b).

Average physical activity was higher among individuals who were not obese (and for women, who were also not overweight), those living in less disadvantaged neighborhoods, with low/no psychological distress and with post school educational qualifications. The lowest

**Table 1. Characteristics of the analytic study sample at entry, by diabetes status.**

| | Respondents N (%) | | Physical Activity† Median (Inter-Quartile Range) | |
|---|---|---|---|---|
| | **Diabetes** | **Non-Diabetes** | **Diabetes** | **Non-Diabetes** |
| **Total** | 362 (5%) | 7338 (95%) | 699 (133–1698) | 849 (300–1898) |
| **Gender** | | | | |
| Men | 186 (51%) | 3096 (42%) | 899 (200–2398) | 999 (316–1998) |
| Women | 176 (49%) | 4241 (58%) | 500 (100–1199) | 799 (300–1764) |
| Missing | 0 (0%) | 1 (0.01%) | | |
| **Education** | | | | |
| No Post School Qualification | 163 (45%) | 2667 (36%) | 399 (100–1199) | 699 (200–1598) |
| Vocational Qualification | 64 (18%) | 1290 (18%) | 799 (258–1723) | 899 (300–1998) |
| Diploma | 41 (11%) | 858 (12%) | 1299 (300–2398) | 899 (400–1998) |
| Bachelor Degree or Higher | 94 (26%) | 2507 (34%) | 999 (325–2473) | 999 (400–1998) |
| Missing | 0 (0%) | 16 (0.2%) | | 416 (92–1232) |
| **BMI** | | | | |
| Underweight | 1 (0.2%) | 104 (1%) | | 599 (200–1824) |
| Normal Weight | 53 (15%) | 2831 (39%) | 799 (300–2398) | 999 (400–1998) |
| Overweight | 115 (32%) | 2858 (39%) | 899 (250–1998) | 899 (300–1948) |
| Class 1 Obesity | 109 (30%) | 1082 (15%) | 599 (100–1399) | 666 (200–1598) |
| Class 2 Obesity | 40 (11%) | 228 (3%) | 483 (125–1698) | 400 (100–1099) |
| Class 3 Obesity | 37 (10%) | 125 (2%) | 500 (0–1199) | 400 (67–1199) |
| Missing | 7 (2%) | 110 (1%) | 599 (225–1049) | 599 (200–1099) |
| **Psychological Distress** | | | | |
| Low/No Distress | 204 (56%) | 4835 (66%) | 799 (200–1998) | 899 (333–1998) |
| Moderate Distress | 85 (24%) | 1606 (22%) | 599 (100–1399) | 799 (266–1698) |
| High Distress | 40 (11%) | 499 (7%) | 549 (100–1474) | 699 (200–1598) |
| Very High Distress | 21 (6%) | 270 (3%) | 400 (100–1199) | 599 (100–1465) |
| Missing | 12 (3%) | 128 (2%) | 553 (29–2198) | 1024 (300–2023) |
| **Annual Gross Household Income (AUD)** | | | | |
| < $25,999 | 73 (20%) | 638 (9%) | 500 (67–1598) | 799 (200–1690) |
| $26,000–51,999 | 79 (22%) | 1234 (17%) | 799 (200–1948) | 699 (200–1698) |
| $52,000–72,799 | 50 (14%) | 968 (13%) | 599 (108–1399) | 799 (266–1798) |
| $72,800–129,999 | 67 (18%) | 1924 (26%) | 749 (200–1948) | 833 (300–1798) |
| ≥ $130,000 | 39 (11%) | 1522 (21%) | 999 (200–2398) | 1199 (450–2199) |
| Don't Know/Don't want to answer | 43 (12%) | 893 (12%) | 799 (167–1399) | 799 (283–1832) |
| Missing | 11 (3%) | 159 (2%) | 500 (350–1324) | 599 (200–1399) |
| **Age Category (years) ‡** | | | | |
| 40–50 | 59 (16%) | 2869 (39%) | 799 (233–1898) | 899 (300–1898) |
| 51–60 | 164 (45%) | 2966 (40%) | 599 (125–1399) | 841 (300–1898) |
| 61–70 | 138 (38%) | 1479 (20%) | 799 (100–1998) | 799 (300–1798) |
| >70 | 0 (0%) | 7 (0.1%) | | 250 (300–1798) |
| Missing | 1 (0.2%) | 17 (0.2%) | | 799 (400–1499) |
| **Neighborhood Disadvantage (Quantiles)** | | | | |
| Quantile 1 (most disadvantaged) | 69 (19%) | 862 (12%) | 599 (100–1798) | 749 (200–1598) |
| Quantile 2 | 88 (24%) | 1284 (17%) | 699 (200–1748) | 699 (233–1598) |
| Quantile 3 | 68 (19%) | 1309 (18%) | 599 (92–1411) | 799 (300–1798) |
| Quantile 4 | 68 (19%) | 1702 (23%) | 749 (183–1523) | 932 (300–1998) |
| Quantile 5 (least disadvantaged) | 53 (15%) | 1956 (27%) | 999 (200–2131) | 999 (400–2098) |

(*Continued*)

**Table 1.** (Continued)

| | Respondents N (%) | | Physical Activity† Median (Inter-Quartile Range) | |
|---|---|---|---|---|
| | **Diabetes** | **Non-Diabetes** | **Diabetes** | **Non-Diabetes** |
| Missing | 16 (4%) | 225 (3%) | 949 (75–2298) | 799 (300–1898) |

† MET.minutes/week. Calculated at first wave with data available within waves 2 to 5. Not reported when <5 people.

‡ Entered as a continuous variable in the models.

levels of physical activity were among obese to very obese women with high levels of psychological distress.

## Physical activity among all respondents

When data from all respondents were included, diabetes was not identified as a predictor of physical activity. BMI, gender, and to a lesser extent, neighborhood disadvantage, psychological distress, income and education remained as important influences on physical activity (Fig 5).

The highest physical activity levels were among men and women who were not obese (and for women, those who were also not overweight), had no/little psychological distress, lived in less disadvantaged neighborhoods, and had post school qualifications.

## Discussion

In this population-based prospective study, people with diabetes engaged in less physical activity than those without diabetes, and this was consistent over time. There were some differences in the factors associated with physical activity between those with diabetes and those without diabetes. The strongest predictor of physical activity among respondents with diabetes was gender, and among those without diabetes was BMI. The strongest predictors of physical activity across all respondents were BMI and gender.

People with diabetes were less active, and had higher BMI than those without diabetes. Higher BMI has previously been associated with type 2 diabetes [32], lower socioeconomic status [33], and lower education [34], making a tree-based analytical approach that allows for complex interactions ideal. A lack of physical activity increases the risk of diabetes and high BMI [35], resulting in a cycle of increasing BMI and less activity leading to increasing BMI and less activity. Other issues which can contribute to low activity among people with diabetes include physical or cognitive disability, depressive symptoms and poor sleep [36].

For people with diabetes, physical activity declined slightly over time as they aged. Other studies have reported declines in health-related quality of life over time among older adults with diabetes [37], and this may result from reduced physical capacity. Diabetes does seem to result in earlier declines in physical function [38], and complications such as neuropathy can cause reduced muscle strength [39] and/or pain [40].

Gender was the strongest predictor of physical activity among respondents with diabetes, and the second strongest predictor of activity among those without diabetes and across the combined sample. Men engaged in more vigorous activity than women, and fewer men than women reported no physical activity each week. Men also engaged in more physical activity than women. This gender difference in physical activity is well-known and exists worldwide [41], with potential reasons including the lack of investment in women's sports and the sociocultural norms resulting in reduced discretionary leisure time for women [42]. Women with

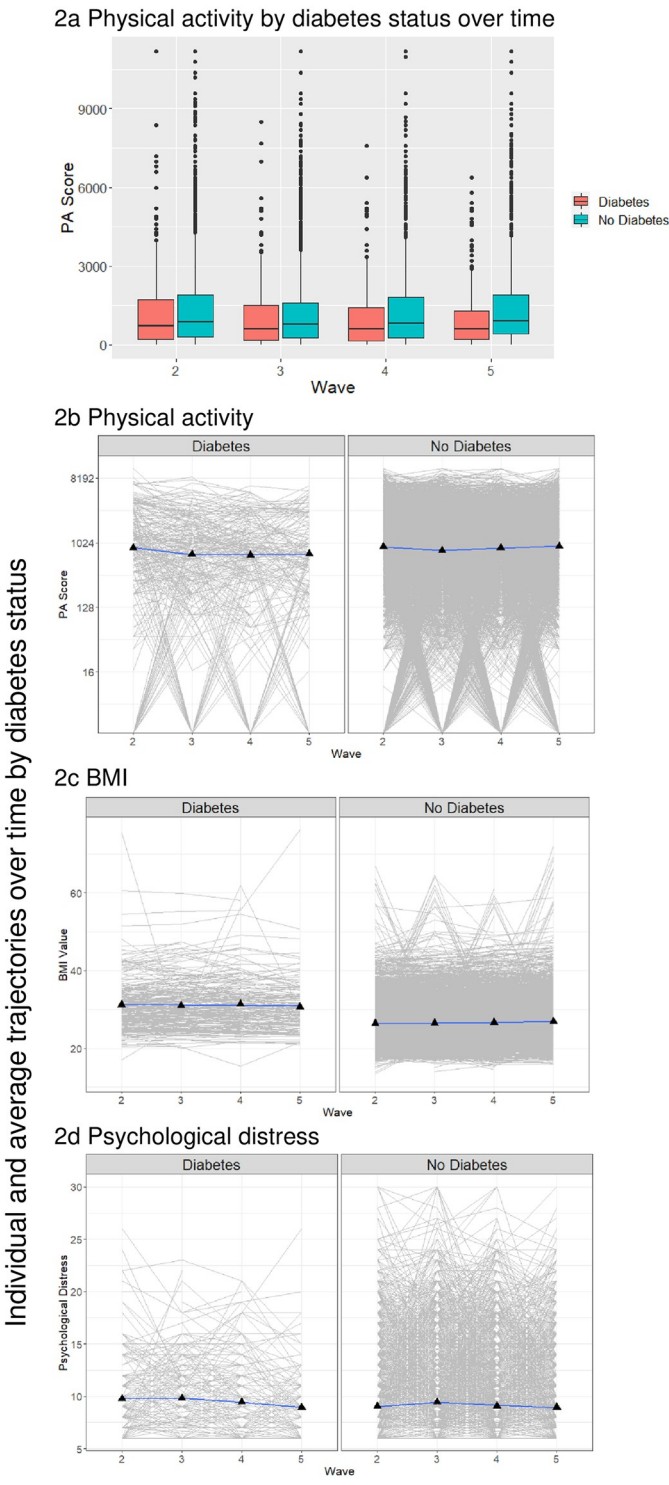

**Fig 2. Changes over time by diabetes status.** Notes: For parts b to d, the black triangles represent the average (mean) value at each wave, the colored line represents the mean trajectory, the grey lines show individual trajectories. Part b has the y-axis shown on a log base 2 scale.

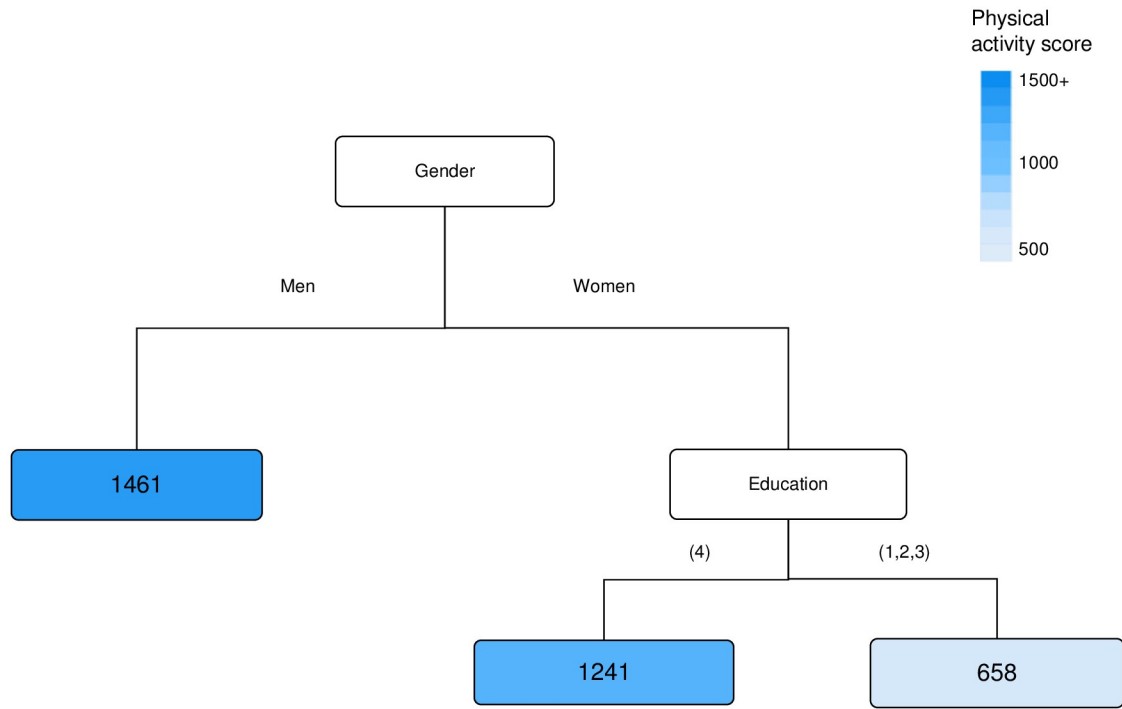

**Fig 3. Factors affecting the physical activity for respondents with diabetes (n = 362).** The terminal nodes show the average physical activity score in MET.min/week. **N**otes: Education (1 = No post school qualification, 2 = Vocational, 3 = Diploma, 4 = Bachelor's or higher), See methods for further details. Predicted Root Mean Square Error per individual and time = 1426 MET mins/week.

diabetes may report lower levels of subjective health along with higher levels of stress and musculoskeletal pain than men, which could also adversely impact physical activity [43].

It was interesting that even the lowest average physical activity level identified in each decision tree (among diabetes respondents: 658 MET minutes/week, non-diabetes: 785, combined: 851) was greater than the 150 mins/week moderate or 75 mins/week vigorous intensity activity (~500 MET minutes/week) recommended for health benefits [15]. However, these values are averages, and large ranges of estimates were observed in the created terminal node categories (S2–S4 Figs). Since the HABITAT cohort was recruited for a study focused on physical activity, survey respondents may have been more active than non-respondents.

This study has certain limitations. Differential attrition occurred in the HABITAT cohort, with a higher proportion of low educated and low income adults leaving the study as time progressed [13]. Self-reported data are vulnerable to bias associated with recall, social desirability and comprehension [44], however the survey questions have been validated in the Australian context. The cohort only included Brisbane residents, and those who moved away were excluded. Findings are likely to be relevant to other metropolitan areas in Australia, but perhaps not rural areas. Information on diet and nutrition, which also impact on BMI, were not captured in the HABITAT study, so could not be considered. Predictors of physical activity may vary by diabetes type, however this study could not differentiate between type 1 and type 2 diabetes. We chose to not include detailed information on neighborhood characteristics (e.g. land use mix, street connectivity), since previous HABITAT research has shown only weak correlation with physical activity among older adults [45].

Strengths of this study include the high response rate over time, plus using data from a study with a unique focus on factors influencing physical activity. Using unbiased REEM tree

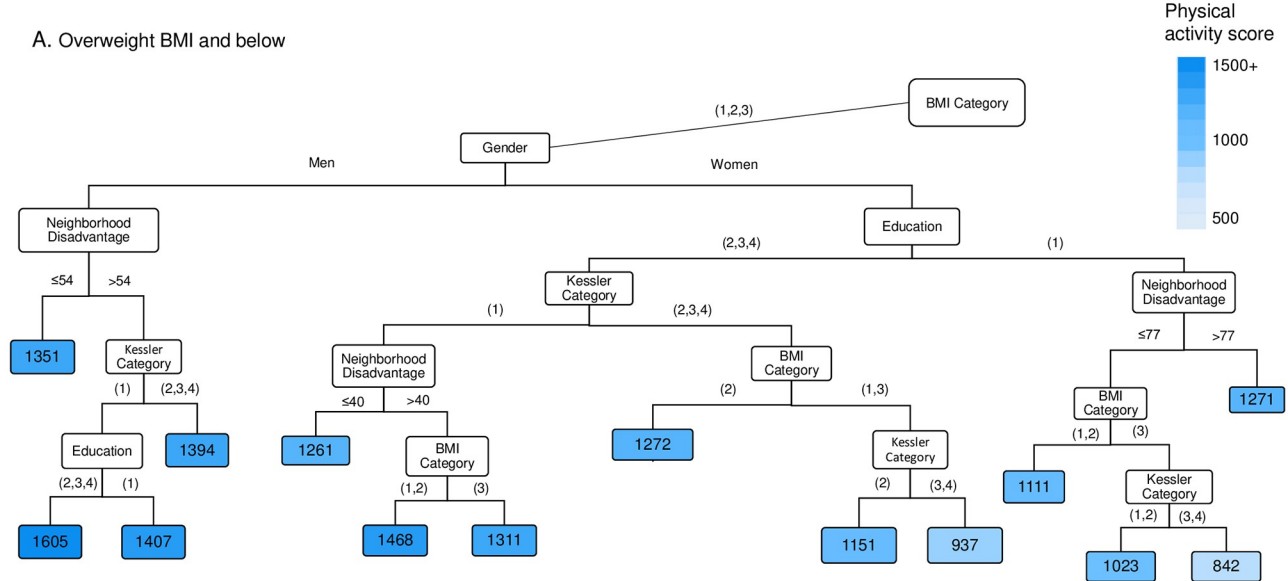

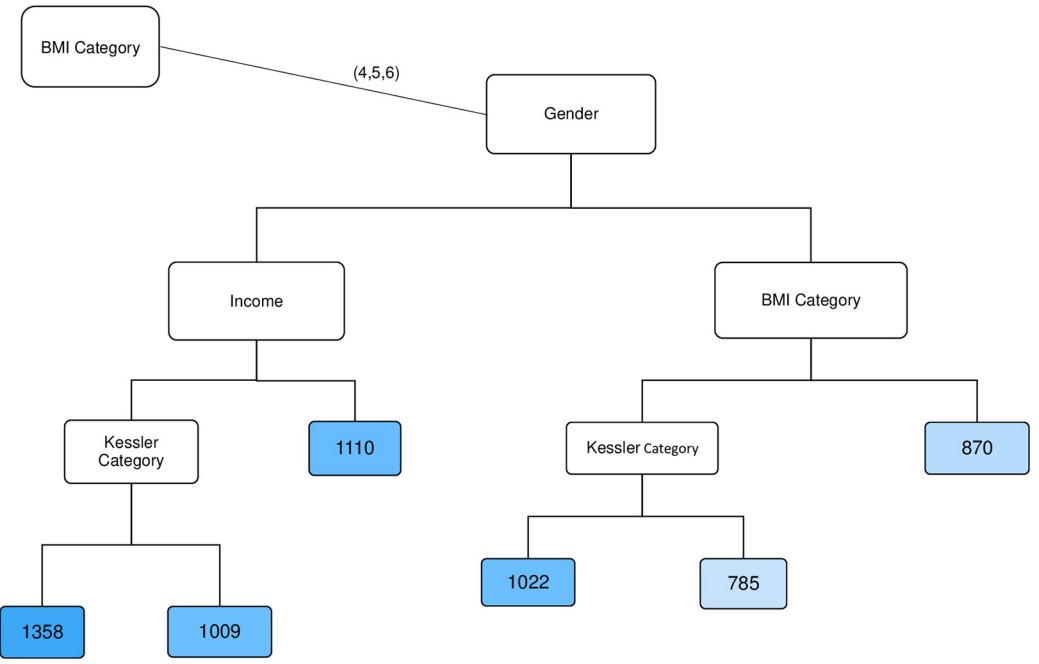

**Fig 4. Factors affecting physical activity for respondents without diabetes (n = 7338).** The terminal nodes show the average physical activity score in MET.min/week. Notes: BMI Category (1 = Underweight, 2 = Normal weight, 3 = Overweight, 4 = Obese 1, 5 = Obese 2, 6 = Obese 3); Education (1 = No post school qualification, 2 = Vocational, 3 = Diploma, 4 = Bachelor's or higher); Kessler Category (1 = Low/none, 2 = Moderate, 3 = High, 4 = Very high); Income (1 = < $25,999, 2 = $26,000–51,999, 3 = $52,000–72,799, 4 = $72,800–129,999, ≥ $130,000, 6 = Don't Know/ Don't want to answer this); See methods for further details. Predicted Root Mean Square Error per individual and time = 1401 MET mins/week.

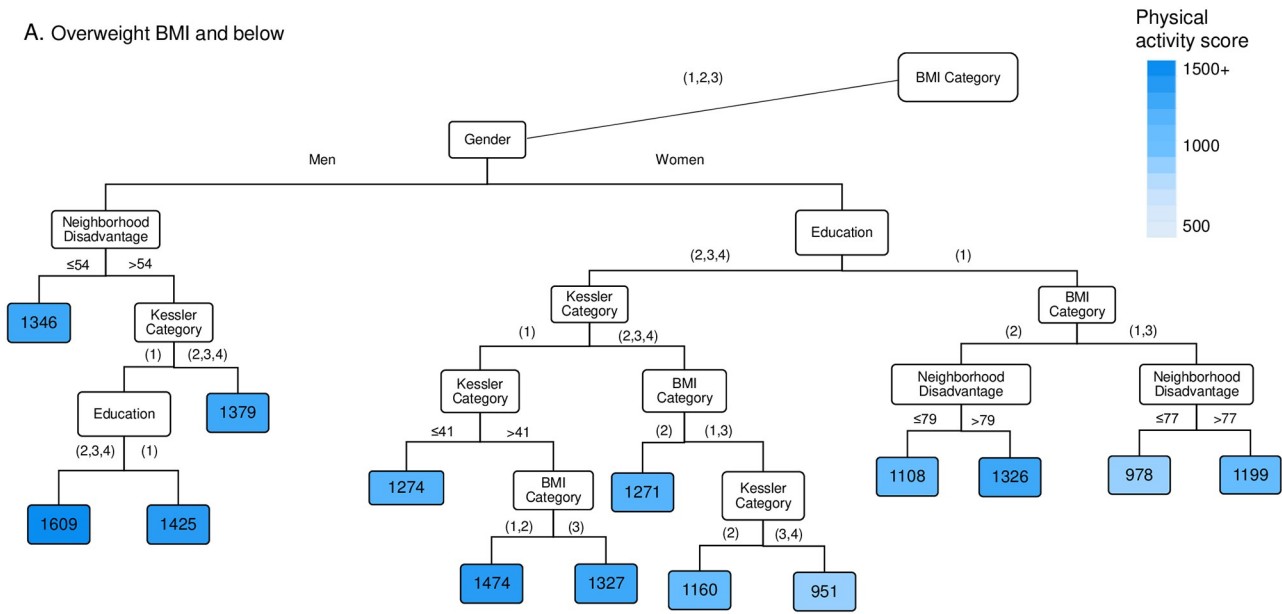

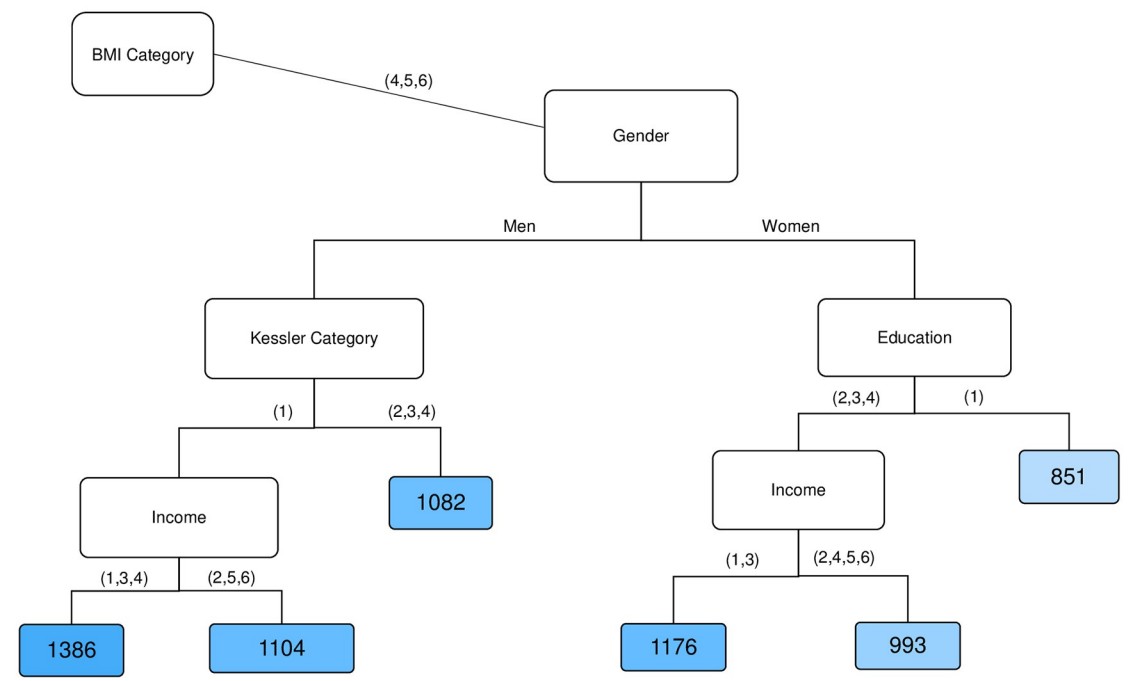

**Fig 5. Factors affecting the physical activity for all respondents (n = 7700).** The terminal nodes show the average physical activity score in MET.min/week. Notes: BMI Category (1 = Underweight, 2 = Normal weight, 3 = Overweight, 4 = Obese 1, 5 = Obese 2, 6 = Obese 3); Education (1 = No post school qualification, 2 = Vocational, 3 = Diploma, 4 = Bachelor's or higher); Kessler Category (1 = Low/none, 2 = Moderate,3 = High, 4 = Very high); Income (1 = < $25,999, 2 = $26,000–51,999, 3 = $52,000–72,799, 4 = $72,800–129,999, ≥$130,000, 6 = Don't Know/ Don't want to answer this). See methods for further details. Predicted Root Mean Square Error per individual and time = 1402 MET mins/week.

methods enabled complex interactions to be explored and could be beneficial in similar studies. Although the first tree model for longitudinal data was published in 1992 [46], these methods remain under-used. A recently released R package ("REEMtree") should help increase uptake of this approach.

## Conclusions

In the current study, mid-aged to older adults with diabetes tended to engage in low levels of physical activity and have high BMI than people without diabetes. However, the strongest influences on physical activity were BMI and gender, not diabetes status. It is vital to promote physical activity among adults, in particular among those with high BMI and women, as well as those with and at high risk of diseases like diabetes.

## Supporting information

**S1 Appendix. Questionnaire excerpt and unbiased RE-EM tree model details.**
(DOCX)

**S1 Table. Studies affirming the relationships shown in S2 Fig.**
(PDF)

**S1 Fig. Directed Acyclic Graph showing influences on diabetes and physical activity.** Notes: Dashed boxes (with associated dashed arrows) indicate the variable was not included in the analysis. See S1 Table for the evidence underlying these relationships.
(TIF)

**S2 Fig. Boxplots for predicted value categories vs physical activity (diabetes respondents).**
(TIF)

**S3 Fig. Boxplots for predicted value categories vs physical activity (non-diabetes respondents).**
(TIF)

**S4 Fig. Boxplots for predicted value categories vs physical activity (all respondents).**
(TIF)

## Acknowledgments

The many helpful suggestions from Gavin Turrell are gratefully acknowledged.

## Author Contributions

**Conceptualization:** Adrian Barnett, Susanna M. Cramb.

**Formal analysis:** H. M. Dumidu A. B. Attanayake.

**Funding acquisition:** Adrian Barnett, Nicola W. Burton, Wendy J. Brown.

**Investigation:** H. M. Dumidu A. B. Attanayake.

**Methodology:** H. M. Dumidu A. B. Attanayake, Adrian Barnett, Nicola W. Burton, Wendy J. Brown, Susanna M. Cramb.

**Supervision:** Nicola W. Burton, Wendy J. Brown, Susanna M. Cramb.

**Validation:** H. M. Dumidu A. B. Attanayake, Susanna M. Cramb.

**Visualization:** H. M. Dumidu A. B. Attanayake, Susanna M. Cramb.

**Writing – original draft:** H. M. Dumidu A. B. Attanayake, Susanna M. Cramb.

**Writing – review & editing:** H. M. Dumidu A. B. Attanayake, Adrian Barnett, Nicola W. Burton, Wendy J. Brown, Susanna M. Cramb.

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
