## [Decision Letter · Decision Letter 0]

18 Apr 2022

PONE-D-22-00526Diabetes and physical activity: a prospective cohort study

PLOS ONE

Dear Dr. Cramb,

Thank you for submitting your manuscript to PLOS ONE. After careful consideration, we feel that it has merit but does not fully meet PLOS ONE’s publication criteria as it currently stands. Therefore, we invite you to submit a revised version of the manuscript that addresses the points raised during the review process.

Apart from comments by the reviewers, the following are additional points to be addressed in revising the manuscript: A more intuitive description of the modeling strategy, specifically, the RE-EM regression tree method, is warranted in that a general audience might find it easier to follow. Further details of the model formulation and supporting theoretical framework for the modeling may be placed in the appendix. Please state the software used.For covariates that can change across waves (including but not limited to psychological distress, BMI), kindly provide more information on their patterns of change, and explain in the discussion section how these might have affected the outcome variable and ultimately the overall results. Explain how missing values were dealt with, and whether missing completely at random can be assumed, and if not, in what way this could bias the study results. 

Other issues to be addressed:

BMI class 3 (line 145) should read "BMI ≥ 40", rather than "BMI > 40". There should not be any gap between adjacent cut-off points for the categories. Similarly for Income categorization (Line 162):  "(5) > $130,000" should be written as "(5) ≥ $130,000".Redo the process chart to make it clearer (see Fig. 1): The lines branching out from the boxes in the left column should be redrawn; they should branch out from the vertical arrows above those boxes. (See example of flowchart, pg 232 - Stephenson T,  Pereira S, Shafran R, Stavola B, Rojas N, McOwat K et al. Physical and mental health 3 months after SARS-CoV-2 infection (long COVID) among adolescents in England (CLoCk): a national matched cohort study. Lancet Child Adolesc Health. 2022; 6: 230–39.)

It is unclear where the 8248 records in relation to missing physical activity come from (see Fig 1: last box in right column, and line 183). This number does not tally with any of the totals in the left column boxes. 

Kindly note that the figure referred to by reviewer 1 regarding high PA scores ("I can see very high scores in fig 1.")  is in fact Figure 2. 

We look forward to receiving your revised manuscript.

Kind regards,

Munn-Sann Lye, MBBS, MPH, DrPH

Academic Editor

PLOS ONE

Journal Requirements:

Reviewers' comments:

Reviewer's Responses to Questions

**Comments to the Author**

1. Is the manuscript technically sound, and do the data support the conclusions?

Reviewer #1: Yes

Reviewer #2: Partly

2. Has the statistical analysis been performed appropriately and rigorously? 

Reviewer #1: I Don't Know

Reviewer #2: I Don't Know

3. Have the authors made all data underlying the findings in their manuscript fully available?

Reviewer #1: Yes

Reviewer #2: No

4. Is the manuscript presented in an intelligible fashion and written in standard English?

Reviewer #1: Yes

Reviewer #2: Yes

5. Review Comments to the Author

Reviewer #1: The study aims to explore whether physical activity differs by diabetes status in mid-aged adults, how this association changes over time, and whether relevant sociodemographic factors and health indicators differ in those with and without diabetes. The authors are using extensive longitudinal data from the HABITAT study, including 4 waves follow-up, and, 7338 non-diabetics, and 362 diabetics respondents. Results are given with interesting visual tree charts/diagrams from the RE-EM modelling.

Although the study is interesting and the results are of novelty, there are few questions and comments that I like to share with the authors.

Main concerns:

- Lack of justification for the modelling

- Difficulties to follow the results

- Self-reported data for physical activity, the calculation is not similar to the reference

Detailed comments:

Outcome: self-reported physical activity

- What was the difference between walking and moderate intensity activity in the questionnaire?

- Why different MET values for walking, moderate and vigorous physical activity were used in the equation of physical activity score compared to the reference 12?

- What is the maximum value for the score ? I can see very high scores in fig 1. Are the high values reliable ?

- I am still a little bit confused whether PA score was used from each time point in the analyses, and what if someone had missing values in one or two time points?

Exposure:

- What do you mean by respondents who changed their diabetes status?

Fig1:

- How was the assistance in daily activities assessed?

- What is the “record”?

Modelling:

- Could you justify more in the text why you have chosen to use the RE-EM modelling instead of other modelling (eg CART)? What is the main purpose of the modelling? Why it is used in this study? What is the advantage of the modelling compared to the other models (more common ways to analyze changes over time)?

- The trees are very interesting and visualize the results well. However, it is very difficult to understand how these trees are done, ie how the modelling gives these results (=trees)? I do not see “age” in these trees, why not?

- What is the role of/What are the terminal nodes? Why multiple terminal nodes/predicted categories was found among non-diabetics but only three terminal nodes among the diabetics, is it because of the disease state or n of the group? Why similar coloring (ie categorization) for PA was used in fig 3, 4 and 5 (I mean, what is the message you are telling/showing, what are the cutpoints)?

Discussion:

- What is meant by “PA declined slightly over time as responders aged”? Which results the authors are referring to?

- How time is treated/taken into account in these analyses? When I read the results/discussion, I still do not understand the role of repeated measurements (expect fig2). The aim given in the end of the introduction includes “change over time”, but I do not find these kind of results.

- Self reported PA is a limitation, however, I am interested if similar analyses could be done with accelerometer measured PA?

- What is the applicability of the RE-EM in the future studies similar to this?

- I would like to see more methodological discussion or justification as the modelling used here is not that common.

Reviewer #2: This is a longitudinal study among a group of adults from Brisbane, Australia who have been followed from 2007 to 2016. The author used the RE-EM (random effects/Expectation-maximization) regression trees and identified factors that affecting physical activity. They reported, the strongest factors influencing physical activity were BMI and gender, but not diabetes status. My main concerns are listed below:

1. If diabetes was diagnosed from one wave, should this individual be considered a patient with diabetes during the following waves no matter what the individual reported during the following waves? If you redefine the diabetes status, I think the sample size will be bigger than 362.

2. (minor point) Line 307, “than those with diabetes”? should be “those without diabetes”?

3. The author stated three aims of the study based on Line 76-78: 1) if PA differs by diabetes status? 2). How this association changes over time; 3) if relevant factors differ in those with and without diabetes? Fig 2 is used to answer Aim 2, but how do we know PA changes differently between diabetes and no diabetes group? From Fig 2, it seems PA did not change much over the 4 waves in either group.

4. I do not really understand the meaning of the PA categories in Fig 3-5 since the exact amount of PA is already shown in the figures. What 1,2,3,4 refers to? Fig 4 and 5 are very hard to see.

6. PLOS authors have the option to publish the peer review history of their article (what does this mean?). If published, this will include your full peer review and any attached files.

Reviewer #1: No

Reviewer #2: No

---

## [Author Response · Author response to Decision Letter 0]

28 Jul 2022

Please find our responses in the "Reponses to Reviewers" pdf uploaded with the manuscript.

---

## [Decision Letter · Decision Letter 1]

18 Aug 2022

PONE-D-22-00526R1Diabetes and physical activity: a prospective cohort studyPLOS ONE

Dear Dr. Cramb,

The comments from Reviewer 1, Reviewer 2 and myself have been appropriately addressed.

Further to my earlier communication, comments by a statistical reviewer (Reviewer 3) have recently been received.

Kindly respond to those comments and submit your revised manuscript by Oct 02 2022 11:59PM. If you will need more time than this to complete your revisions, please reply to this message or contact the journal office at plosone@plos.org. Please include the following items when submitting your revised manuscript:

A marked-up copy of your manuscript that highlights changes made to the original version. You should upload this as a separate file labeled 'Revised Manuscript with Track Changes'.An unmarked version of your revised paper without tracked changes. You should upload this as a separate file labeled 'Manuscript'.

We look forward to receiving your revised manuscript.

Kind regards,

Munn-Sann Lye, MBBS, MPH, DrPH

Academic Editor

PLOS ONE

Journal Requirements:

Reviewers' comments:

Reviewer's Responses to Questions

**Comments to the Author**

1. If the authors have adequately addressed your comments raised in a previous round of review and you feel that this manuscript is now acceptable for publication, you may indicate that here to bypass the “Comments to the Author” section, enter your conflict of interest statement in the “Confidential to Editor” section, and submit your "Accept" recommendation.

Reviewer #1: All comments have been addressed

Reviewer #2: All comments have been addressed

Reviewer #3: (No Response)

2. Is the manuscript technically sound, and do the data support the conclusions?

Reviewer #1: (No Response)

Reviewer #2: Yes

Reviewer #3: Yes

3. Has the statistical analysis been performed appropriately and rigorously? 

Reviewer #1: (No Response)

Reviewer #2: Yes

Reviewer #3: Yes

4. Have the authors made all data underlying the findings in their manuscript fully available?

Reviewer #1: (No Response)

Reviewer #2: Yes

Reviewer #3: No

5. Is the manuscript presented in an intelligible fashion and written in standard English?

Reviewer #1: (No Response)

Reviewer #2: Yes

Reviewer #3: Yes

6. Review Comments to the Author

Reviewer #1: (No Response)

Reviewer #2: The authors have addressed my comments adequately. I am satisfied and no additional questions. Great job.

Reviewer #3: A research study was conducted which aimed to describe changes in physical activity over time and compare physical activity based on diabetes status. At baseline, participants with diabetes had a lower physical activity levels than those without diabetes. Random effects/Expectation-maximization regression trees showed that the strongest factors influencing physical activity were BMI and gender, not diabetes status.

Minor revisions:

1- Abstract: Provide p-values to support the following statement. “At study entry, those with diabetes had a higher median age of 58 years and a lower median physical activity of 699 MET.min/week than people without diabetes (53 years and 849 MET.min/week).”

2- Lines 253-262: Provide measures of dispersion of the median ages and physical activity scores.

3- The standard statistical term for average is mean.

4- Table 1: Typically missing categories are presented last in each listing, and they are not included in the percentage calculation.

7. PLOS authors have the option to publish the peer review history of their article (what does this mean?). If published, this will include your full peer review and any attached files.

Reviewer #1: No

Reviewer #2: No

Reviewer #3: No

---

## [Decision Letter · Decision Letter 2]

13 Oct 2022

Diabetes and physical activity: a prospective cohort study

PONE-D-22-00526R2

Dear Dr. Cramb,

We’re pleased to inform you that your manuscript has been judged scientifically suitable for publication and will be formally accepted for publication once it meets all outstanding technical requirements.

Kind regards,

Munn-Sann Lye, MBBS, MPH, DrPH

Academic Editor

PLOS ONE

Additional Editor Comments (optional):

Reviewers' comments:

Reviewer's Responses to Questions

**Comments to the Author**

1. If the authors have adequately addressed your comments raised in a previous round of review and you feel that this manuscript is now acceptable for publication, you may indicate that here to bypass the “Comments to the Author” section, enter your conflict of interest statement in the “Confidential to Editor” section, and submit your "Accept" recommendation.

Reviewer #3: All comments have been addressed

2. Is the manuscript technically sound, and do the data support the conclusions?

Reviewer #3: (No Response)

3. Has the statistical analysis been performed appropriately and rigorously? 

Reviewer #3: (No Response)

4. Have the authors made all data underlying the findings in their manuscript fully available?

Reviewer #3: (No Response)

5. Is the manuscript presented in an intelligible fashion and written in standard English?

Reviewer #3: (No Response)

6. Review Comments to the Author

Reviewer #3: (No Response)

7. PLOS authors have the option to publish the peer review history of their article (what does this mean?). If published, this will include your full peer review and any attached files.

Reviewer #3: No

---

## [Editor Report · Acceptance letter]

17 Oct 2022

PONE-D-22-00526R2 

Diabetes and physical activity: a prospective cohort study 

Dear Dr. Cramb:

I'm pleased to inform you that your manuscript has been deemed suitable for publication in PLOS ONE. Congratulations! Your manuscript is now with our production department. 

Kind regards, 

on behalf of

Professor Munn-Sann Lye 

Academic Editor

PLOS ONE